# Growth Performance, Metabolomics, and Microbiome Responses of Weaned Pigs Fed Diets Containing Growth-Promoting Antibiotics and Various Feed Additives

**DOI:** 10.3390/ani14010060

**Published:** 2023-12-23

**Authors:** Michaela P. Trudeau, Wes Mosher, Huyen Tran, Brenda de Rodas, Theodore P. Karnezos, Pedro E. Urriola, Andres Gomez, Milena Saqui-Salces, Chi Chen, Gerald C. Shurson

**Affiliations:** 1Department of Animal Science, University of Minnesota, St. Paul, MN 55108, USA; mmetz@landolakes.com (M.P.T.); urrio001@umn.edu (P.E.U.); gomeza@umn.edu (A.G.); msaquisa@umn.edu (M.S.-S.); 2Department of Food Science and Nutrition, University of Minnesota, St. Paul, MN 55108, USA; moshe096@umn.edu (W.M.); chichen@umn.edu (C.C.); 3Purina Animal Nutrition, Gray Summit, MO 63039, USA; httran@landolakes.com (H.T.); bderodas@landolakes.com (B.d.R.); tpkarnezos@landolakes.com (T.P.K.)

**Keywords:** antibiotic growth promoters, feed additives, growth performance, metabolomics, microbiome, weaned pigs

## Abstract

**Simple Summary:**

Understanding the biological mechanisms associated with growth performance improvements that may occur when adding various feed additives to diets of weaned pigs fed is essential for their strategic and optimal use. The objective of this study was to evaluate the growth performance, metabolic profiles, and intestinal microbiome composition of nursery pigs fed various feed additives and to determine potential biological mechanisms associated with growth promotion. The growth performance responses of high-health weaned pigs (20 days of age) were determined in three separate 42 day experiments when fed antibiotics (chlortetracycline and sulfamethazine; PC), herbal blends, turmeric, garlic, bitter orange extract, sweet orange extract, volatile and semi-volatile milk-derived substances, yeast nucleotide, and cell wall products, compared with feeding a non-supplemented diet (NC). None of the feed additives except antibiotics significantly improved growth performance, compared with feeding NC. Furthermore, none of the dietary treatments affected the metabolome and microbiome profiles or alpha and beta microbiome diversity in the ileum and cecum. However, some additives, such as herbal blends and garlic, increased the relative abundance of certain bacteria genera in the ileal and cecal microbiome that may provide some protection for weaned pigs when experiencing disease challenge.

**Abstract:**

The objective of this study was to determine the potential biological mechanisms of improved growth performance associated with potential changes in the metabolic profiles and intestinal microbiome composition of weaned pigs fed various feed additives. Three separate 42 day experiments were conducted to evaluate the following dietary treatments: chlortetracycline and sulfamethazine (PC), herbal blends, turmeric, garlic, bitter orange extract, sweet orange extract, volatile and semi-volatile milk-derived substances, yeast nucleotide, and cell wall products, compared with feeding a non-supplemented diet (NC). In all three experiments, only pigs fed PC had improved (*p* < 0.05) ADG and ADFI compared with pigs fed NC. No differences in metabolome and microbiome responses were observed between feed additive treatments and NC. None of the feed additives affected alpha or beta microbiome diversity in the ileum and cecum, but the abundance of specific bacterial taxa was affected by some dietary treatments. Except for feeding antibiotics, none of the other feed additives were effective in improving growth performance or significantly altering the metabolomic profiles, but some additives (e.g., herbal blends and garlic) increased (*p* < 0.05) the relative abundance of potentially protective bacterial genera that may be beneficial during disease challenge in weaned pigs.

## 1. Introduction

Many commercial feed additives are marketed with the goal of improving the health and growth performance of nursery pigs. Poor growth performance after weaning is often associated with low feed intake and chronic intestinal disorders, which many of these products are intended to ameliorate [1]. Despite the health and growth performance benefits provided by many feed additives, there is limited information on the mechanisms by which they affect the biological systems of pigs, including metabolism and the gastrointestinal microbiome. Understanding potential metabolic and gastrointestinal microbiome changes associated with the growth and health improvements of feed additives is necessary for ensuring that feed additives are strategically used in situations where they will have the greatest likelihood of consistently providing beneficial effects.

Although some studies have shown growth performance improvements from feeding herbal blends, phytogenic extracts, yeast-based products, and milk-derived substances, responses have been inconsistent [2,3,4,5,6,7,8,9]. Herbal blends are botanical extracts derived from multiple types of herbal plants, while phytogenic extracts are compounds derived from single plants, spices, and fruits [4]. Both herbal blends and phytogenic extracts have antimicrobial, anti-inflammatory, and antioxidant capabilities that are mostly attributed to phenolic compounds, flavonoids, and carotenoids [4,9,10,11,12].

Similarly, a few studies have shown that the addition of yeast products to weaned pig diets improved average daily gain (ADG) and gain:feed (G:F), but the magnitude of response is variable [13,14,15]. Differences in the relative growth responses of pigs to these yeast additives could be due to a variety of yeast characteristics including prebiotic and probiotic effects [16,17], mannan oligosaccharide content [18,19,20], or nucleotide concentrations [21,22,23]. Unfortunately, little is known about how diet composition, animal health, and environmental conditions in production facilities may impact the efficacy of these products.

Milk-derived substances are produced during milk fermentation and commonly include butyric and acetic acid [24]. It is unclear if these milk-derived substances improve the taste of the diets and subsequent feed intake in weaned pigs, or whether adding organic acids, including butyric acid and acetic acid, to diets improves the growth performance by improving nutrient digestibility and reducing intestinal pathogen infections [25,26,27]. Butyric acid has been shown to provide energy to the intestinal epithelium, which is important during recovery from intestinal damage during weaning [28,29]. In addition, butyrate serves as a transcription factor promoting cell proliferation and differentiation in the intestine, along with regulating epithelial inflammation and tolerance to antigens through the production of cytokines and induction of tolerogenic dendritic cells [30]. However, these responses have not been explored or shown when feeding milk-derived substances to weaned pigs.

The inconsistency of growth responses when weaned pigs are fed diets supplemented with herbal blends, phytogenic extracts, yeast products, and milk-derived substances may be due to differences in the types of active compounds and their concentrations in these additives. By evaluating potential changes in the metabolic profiles and intestinal microbiome composition in pigs fed diets containing these additives, patterns may be identified that can provide insight for determining their mechanisms for promoting growth and health. For this reason, we hypothesized that these feed additives will improve the growth performance of nursery pigs through modulation of the intestinal microbiome and metabolome. Therefore, the objective of this study was to determine if feeding diets containing recommended amounts of herbal blends, phytogenic extracts, yeast products, or milk-derived substances to weaned pigs improves growth performance and whether changes in the metabolome and microbiome at intestinal and systemic (serum) levels may be associated with these improvements.

## 2. Materials and Methods

Three feeding experiments and sample collection were conducted at the Purina Animal Nutrition Research Farm (Gray Summit, MO, USA) and followed experimental protocols approved by the Institutional Animal Care and Use Committee (IACUC) which met all federal requirements for responsible conduct and ethical treatment of experimental animals. Experiment (EXP) 1 was conducted from 11 June to 26 July 2018 and followed IACUC approved study protocol PS1109 which included IACUC approved SOP #SRU-002 (nursery pigs; approved 2 February 2012), SOP #SRU-009 (animal health; approved 2 February 2012), and SOP # SRU-015 (pig weighing procedures; approved 19 April 2010). Experiment 2 was conducted from 16 July to 30 August 2018 and followed IACUC approved study protocol WF019 which included IACUC approved SOP #SRU-002, SOP #SRU-009, and SOP # SRU-015. Experiment 3 was conducted from 20 August to 4 October 2018 and followed IACUC approved study protocol PS1110 which included IACUC approved SOP #SRU-002, SOP #SRU-009, and SOP # SRU-015. One on-site veterinarian and one external veterinarian were responsible for monitoring compliance of following the IACUC approved protocols by Purina Animal Nutrition research personnel. All sample analyses for metabolome and microbiome comparisons were conducted at the University of Minnesota (St. Paul, MN, USA). The methodologies used in the current study followed the same procedures for comparing growth performance, gut microbiome, and metabolome responses among experimental facilities previously described [31].

### 2.1. Animals, Housing, and Experimental Design

Three separate 42 day nursery pig feeding experiments (EXP) were conducted, where EXP 1 and 2 were performed in the same nursery facility, and EXP 3 was conducted in a wean-to-finish facility located on the same research farm. The nursery facility was environmentally controlled and included plastic flooring, five-hole plastic feeders, and nipple waterers. The wean-to-finish facility included slatted concrete flooring, five-hole metal feeders, and cup waterers. Room temperature in both facilities was maintained at about 30 °C during the first week after weaning and subsequently decreased by 1.8 °C per week during the 6-week nursery period.

Experiment 1, 2, and 3 included 504, 516, and 520 newly weaned mixed-sex pigs (20 days of age), respectively. The pigs in all EXP were from the same genetic line (PIC Camborough × PIC 337; Hendersonville, TN, USA), had an average initial body weight (BW) of 6.5 kg, and were blocked by initial BW and sex and randomly assigned to treatments to provide at least 8 pens/treatment. In EXP 1 and 2, 8 to 9 pigs were placed per pen to provide a stocking density of 0.36 to 0.41 m^2^/pig, while in EXP 3, 12 to 14 pigs were placed in each pen to provide a stocking density of 0.58 to 0.63 m^2^/pig.

All pigs were vaccinated for *Streptococcus suis* and *Mycoplasma hyorhinis* (Autogenous Bacterin, Phibro Animal Health, Teaneck, NJ, USA) at 5 to 7 days of age (booster at weaning), for *Haemophilus parasuis* (ParaSail, Newport Laboratories, Worthington, MN, USA) and *Salmonella typhimurium* (Enterisol-Salmonella T/C; BI, St. Joseph, MO, USA) 7 days prior to weaning, and for Circovirus Type 2 (Fostera PCV Chimera, Zoetis, Charles City, IA, USA) at weaning. Throughout all three EXP, pigs were monitored daily for health status for the duration of each EXP.

All pigs and feeders in each pen were weighed on day 0, 10, 21, and 42. Body weight data were used to calculate ADG, and feeder weight along with recorded amounts of each diet added to feeders during each feeding period were used to calculate feed disappearance and average daily feed intake (ADFI). Pen BW gain and feed disappearance during each feeding period were used to calculate G:F.

### 2.2. Dietary Treatments

All diets were formulated to meet or exceed the NRC (2012) recommended nutrient requirements of nursery pigs. Phase 1 diets were fed from day 0 to 10, phase 2 diets were fed from day 10 to 21, and phase 3 diets were fed from day 21 to 42 post-weaning. The nursery diets were semi-complex corn soy-based diets with a total lysine of 1.58%, 1.50%, and 1.41% in phases 1, 2, and 3, respectively. The metabolizable energy of the diets was 3405, 3355, and 3355 kcal/kg. Copper was added to all nursery diets at 175 ppm. Pharmaceutical levels of Zinc were added into the phase 1 diet at 3000 ppm and in the phase 2 at 2000 ppm. In the phase 3 diet, Zinc was lowered to 150 ppm. All diets were manufactured and pelleted at the Purina Animal Nutrition Research Feed Manufacturing Facility. Pens were assigned randomly to dietary treatments (Table 1). The positive control (PC) diet contained 0.5% Aureomix 10-10S (Zoetis; Charles City, IA, USA) to provide 0.01% chlortetracycline and 0.01% sulfamethazine. Different proprietary herbal blend products PHY01, PHY02, and PHY03 were added to diets at manufacturer-recommended inclusion rates of 0.03%, 0.1%, and 0.02%, respectively. Dietary treatments containing phytogenic extracts containing 0.01% turmeric (TUM), 0.015% garlic (GAR), 0.03% bitter orange extract (BOE), and sweet orange extract (SOE) were added at inclusion rates of 0.037 to 0.018% of the diet, depending on dietary phase. Proprietary volatile (VM01) and semi-volatile (VSM02) milk-derived substances were added to diets at a rate of 0.05% and 0.03%, respectively, based on manufacturers recommendations. Yeast-derived additives including yeast nucleotide products (YN01 and YN02) and a yeast cell wall product (YC03) were added to diets at inclusion rates of 0.1 to 0.05%, depending on the dietary phase. All pigs were provided with ad libitum access to feed and water throughout each EXP.

### 2.3. Chemometric Analysis of Additives

All feed additives were analyzed for chemical composition in original and microwave digested forms. Chemometric analysis of the original forms involved extraction using 50% acetonitrile followed by analyzing these extracts using three different LC-MS methods: positive mode, negative mode, and dansylation, while analysis of the digested forms involved hydrolyzing each additive using HCl followed by microwave digestion. The hydrolysates were dried with N_2_ and reconstituted using 50% acetonitrile. Data were then subjected to a multivariate analysis, and the major chemical components found in each additive were identified using the Metlin (metlin.scripps.edu; accessed on 12 November 2019) database search.

### 2.4. Statistical Analysis of Growth Performance Data

The minimum number of replicates per dietary treatment in each EXP was determined by calculating the necessary sample size to achieve statistical significance at *p* < 0.05 at a power of 0.80 using data from previous experiments as inputs and G*Power 3.1 (Kiel University, Kiel, Germany). Growth performance data (BW, ADG, ADFI, and G:F) were analyzed for normal distribution and absence of outliers using the UNIVARIATE procedure of SAS (SAS Institute; Cary, NC, USA). Pen was the experimental unit. Experimental data were analyzed as a randomized complete block design using the GLIMMIX procedure of SAS with time as a repeated measure and Autoregressive 1 variance structure. Replicate was included as a random effect and dietary treatments were fixed effects in the model. Significant differences were declared at *p* ≤ 0.05.

### 2.5. Sample Collection

One pig with BW closest to the median BW in each pen was selected for blood sample collection and subsequently euthanized for digesta content sample collection on day 42 of the EXP. Blood samples were collected via venipuncture of the jugular vein and samples were collected in Vacutainer^®^ blood collection tubes (BD; Franklin Lakes, NJ, USA) and then centrifuged at 2000× *g* for 15 min at 4 °C. The serum was then aliquoted and stored at −80 °C. Pigs were subsequently euthanized using CO_2_ gas and exsanguination, followed by removing the entire intestinal tracts, placing them on a sterile surface, and using sterilized utensils to collect digesta samples. Approximately 1.5 mL of ileum contents was collected 30 cm proximal to the ileocecal junction, and 1.5 mL of cecal content was collected from the lateral side of the cecum. Each sample was then snap-frozen in liquid nitrogen and stored at −80 °C.

### 2.6. Metabolomics Analyses

The details of the metabolomics methods have been described previously [31]. Individual metabolite concentrations were calculated using the ratio between the peak area of metabolite and the peak area of internal standard and fitting with a standard curve using QuanLynx software version 4.1 (Waters Corp, Milford, MA, USA). The aov function from the statistic package in Rstudio was used to compare metabolite concentrations across treatments (Posit PBC; Boston, MA, USA; version 3.5.3) [32]. Pairwise comparisons against the negative control were completed using a Wilcoxon test. For analysis of the untargeted metabolites, the processed data matrix was analyzed by unsupervised principal components analysis (PCA) using Rstudio software version 3.5.3. The weighted Bray–Curtis matrix was created using the vegan package in Rstudio [33] and the adonis function in the vagan package was used to calculate PERMANOVA values [34], which was used to quantify separation between treatment groups. Metabolite indicator values were calculated using the labdsv package in Rstudio [35]. Indicator values are used to quantify the occurrence and amount of a given metabolite in all samples from a specific treatment. Indicator values close to 1 indicate that the metabolite is present in multiple samples of that treatment and are in high abundance compared to other treatment groups. A high indicator value is generally considered to be greater than 0.8, and only indicator values greater than 0.8 were reported in the results of this study. The chemical identities of compounds of interest were determined by accurate mass measurement, elemental composition analysis, followed by a database search using the Metlin search engine (metlin.scripps.edu; accessed 12 November 2019).

### 2.7. Microbiome Analyses

All ileum (n = 150) and cecal (n = 150) samples were submitted to the University of Minnesota Genomics Center for DNA extraction and sequencing of the 16S rRNA V4 region. The DNeasy PowerSoil DNA extraction kit (Qiagen, Hilden, Germany) was used for DNA extraction by following manufacturer protocols. Library prep was completed using the dual-indexing method [36]. Marker gene sequencing was then completed using the Illumina MiSeq Next Generation platform with a targeted average sequencing depth of 100,000 reads per sample.

Amplicon Sequence Variants (ASV) were generated from raw Illumina sequence reads using various open-source software including cutadapt, fastx, and qiime2 [37,38,39]. Primers and empty lines were removed using the default parameters of the cutadapt program. The sequences were filtered using a quality score criteria of Q = 30 and all other parameters set to default settings. The paired ends were then merged using bbmap merger and singletons were discarded [40]. Sequences were demultiplexed and processed through the dada2 plugin of qiime2 to identify ASVs [41]. Taxonomic assignments of the ASVs were completed using a pre-trained classifier and the greengenes database [42]. After data processing, 72% high-quality reads were retained.

Alpha diversity (rarefied richness and Simpson index) and beta diversity (Bray–Curtis-based principal coordinate analyses and permutational multivariate analyses of variance) metrics were determined using the vegan package of RStudio [43]. Species indicator analyses were calculated using the labdsv R package [44].

## 3. Results

### 3.1. Chemical Composition of Feed Additives

Based on chemometrics analysis of the products evaluated in EXP 1, all of the herbal blends (PHY01, PHY02, and PHY03) had relatively high concentrations of choline, glycine, and pipecolic acid. The PHY01 blend had a relatively high concentration of piperine, whereas PHY02 contained curcumin. Curcumin was also identified as a unique component in the phytogenic extract TUM, while GAR contained ibervirin. The BOE and SOE phytogenic extracts were distinguished from other additives due to a greater relative abundance of flavonoids. Quinic acid was identified as a main constituent of the VSM02 product, while the yeast products contained greater relative abundance of free amino acids and dipeptides compared with the other feed additives evaluated (see Appendix A).

### 3.2. Growth Performance Responses

#### 3.2.1. EXP 1

Dietary treatments consisted of NC, PC, PHY01, PHY02, PHY03, TUM, and GAR in this EXP. Two pigs were removed from the PC treatment due to a *Streptococcus suis* infection, and one pig was removed from the NC treatment because of lameness during the EXP. There were no differences in BW, ADG, or ADFI among treatment groups (Table 2). However, there was an interaction (*p <* 0.05) between time and treatment for G:F, indicating that these dietary treatments had a variable effect on gain efficiency depending on the age of the pigs. Gain:feed was less (*p* < 0.05) for pigs fed PC during the first 10 days post-weaning compared with those fed NC, PHY01, TUM, and GAR treatments. During phase 2, (days 10 to 21), pigs fed PC had greater (*p* < 0.05) G:F compared with pigs fed PHY01.

#### 3.2.2. EXP 2

Dietary treatments consisted of NC, PC, BOE, SOE, VM01, and VSM02 in this EXP. One pig in the PC treatment and one pig in the NC treatment were removed because of lameness during the EXP. There were no differences between dietary treatments for BW, ADG, ADFI, or G:F during phase 1 (0 to 10 days postweaning) (Table 3). However, during phase 2 (days 10 to 21 post-weaning), pigs fed PC had a greater (*p* < 0.05) BW than those fed SOE and had a greater (*p* < 0.05) ADG than pigs fed NC, BOE, SOE, and VSM02 (Table 3). Pigs fed PC diets had a greater (*p* < 0.05) ADFI during phase 2 compared with pigs fed SOE and VSM02. During phase 3 (day 22 to day 42 post-weaning), pigs fed PC had a greater (*p* < 0.05) BW than pigs fed NC, BOE, SOE, and VSM02.

#### 3.2.3. EXP 3

Dietary treatments consisted of NC, PC, YN01, YN02, and YC03 in this EXP. One pig was removed from the PC and NC treatments due to a *Streptococcus suis* infection during the EXP. There was a treatment × time interaction for BW, indicating that dietary treatment had a variable effect on growth based on the age of pigs (Table 4). Pigs fed PC were heavier (*p* < 0.05) on day 42 compared with those fed NC, YN01, YN02, and YC03. However, there were no differences in ADG or ADFI among dietary treatments during the entire 42 day feeding period. Pigs fed PC had a greater (*p* < 0.05) G:F compared with pigs fed YN02 only during the phase 1 (days 0 to 10 post-weaning) feeding period. None of the yeast products evaluated affected the growth performance of weaned pigs compared with feeding the NC diets.

### 3.3. Targeted Metabolomics

#### 3.3.1. Serum

There were no differences in serum amino acid concentrations among any of the dietary treatments in EXP 1 (see Appendix A) and 2 (see Appendix A). In EXP 3, pigs fed PC had greater (*p* < 0.05) serum lysine concentration compared with pigs fed NC and diets containing yeast products (see Appendix A). The concentration of serine was also greater (*p* < 0.05) in pigs fed YC03 compared with those fed NC (see Appendix A). Serum metabolite biomarkers were identified for each treatment. Sulfadimidine, which is an antibiotic metabolite found in pigs fed PC diets, was the only biomarker that had a high indicator value (see Appendix A).

#### 3.3.2. Cecal Contents

There were no differences in short-chain fatty acid concentrations in cecal contents among the dietary treatments in EXP 1. There were a few trends for differences in cecal content concentrations of three amino acids for pigs fed GAR, one amino acid for pigs fed PHY03, and one bile acid for pigs fed PHY02, compared with those from pigs fed NC diets (see Appendix A). In EXP 2, there were no differences in amino acid concentrations in cecal contents between pigs fed PC, BOE, SOE, VM01, and VSM02 and those fed NC, but pigs fed VM01 had less acetic, propionic, and butryric acid than those fed NC diets (see Appendix A). Only one bile acid was increased in the cecal contents of pigs fed PC compared with those fed NC diets (see Appendix A). In EXP 3, pigs fed PC diets had greater concentrations of glutamic acid, histidine, leucine-isoleucine, phenylalanine, tyrosine, and cholic acid than those fed NC and yeast product treatments (see Appendix A). Similar to the findings for the serum metabolites, sulfadimidine and chlortetracycline metabolites were identified in the PC samples of cecal contents (Appendix A). In addition, metabutamine was identified as a biomarker for PHY02, and butyramide was a biomarker for PHY03. Ibervirin was identified as a biomarker in the cecal contents of pigs fed GAR (Appendix A), which was also identified as a component of the GAR product from the chemometrics results. Among the yeast-based product dietary treatments, only pigs fed YN02 had a biomarker (phloionolic acid) with a high indicator value.

### 3.4. Microbiome Profiles

Despite the addition of antibiotics and various feed additives to nursery pig diets, there were no differences among treatments in either the Simpson alpha diversity index, rarefied richness, or beta diversity in the cecal or ileal content samples (Figure 1, Figure 2, Figure 3 and Figure 4). However, specific taxa were identified in the cecal contents for bacterial strains that were impacted by the dietary treatments. Feeding PHY01, PHY02, PHY03, and GAR increased the relative abundance of ASVs affiliated to *Lactobacillus* compared with pigs fed NC (Table 5). Pigs fed diets containing YN01 had increased relative abundance of *Megasphaera* and *Prevotella stercorea,* compared with pigs fed NC (Table 5). In the same EXP, *Streptomycetaceae* was identified as a significant biomarker of PC because it was only present in the intestinal tract when the antibiotics were fed to pigs (Table 5).

## 4. Discussion

### 4.1. Active Compounds in Feed Additives

Based on the chemometric analysis, PHY01 was unique from the other herbal blends because it contained high concentrations of piperine, which is an alkaloid found in black pepper. Although research on this compound is limited, feed additives containing piperine have been shown to improve ADG and ADFI in nursery pigs [6]. The herbal blend PHY02 and phytogenic extract TUM contained relatively high concentrations of curcumin. Curcumin is a polyphenolic compound typically found in turmeric, which is a common spice used in foods [45]. Turmeric has been reported to provide antioxidant, anti-inflammatory, and anti-microbial effects when fed to rodents [45,46,47]. Previous feeding trials have shown improved growth of nursery pigs when fed diets containing turmeric powder, but it is unclear if this effect was due to curcumin alone or if it was attributed to the other plant substances in turmeric including tumerones, sesquiterpenes, stigmasterole, β-sitosterole, and cholesteroland [48,49]. All three herbal blends (PHY01, PHY02, and PHY03) had relatively high concentrations of quinic acid, which has been shown to increase tryptophan and nicotinamide production in the gastrointestinal tract and enhance DNA repair and immune function via NF-kB inhibition [50]. However, in this study, there were no differences in tryptophan concentrations in the cecal contents when herbal blends were included in the diet compared with pigs fed NC. The herbal blend PHY03 was unique compared with other herbal blends because of its relatively high concentrations of betaine, which is a derivative of glycine and functions as a methyl group donor. Betaine has been shown to have osmotic properties that may be beneficial to the intestinal epithelium of pigs [46]. When betaine was added at a rate of 0.15% in the diet, ADG increased up to 15% in pigs [51]. Despite these potential advantages, it is likely that the small amount of betaine in PHY03, and the even lower concentration in the PHY03 diet when added at an inclusion rate of 0.02%, was not great enough to have a measurable effect on the growth performance.

Bitter orange extract and sweet orange extract were similar in chemical composition and contained relatively high concentrations of flavonoids including equol, nobiletin, and tangeritin. Flavonoids may be beneficial in swine diets due to their antioxidant, anti-inflammatory, anti-microbial, and anti-carcinogenic properties, and have been shown to have an inhibitory effect against viruses in humans [52,53,54]. In a previous study, when citrus plant extracts were included in weaned pig diets at a rate of 0.1% and 0.2%, pigs had greater ADG regardless of inclusion rate, compared with pigs fed negative control diets [7]. In contrast, the results from another study showed that the addition of 0.3% Chinese herbal blends, containing high concentrations of flavonoids and polyphenols, to weaned pig diets resulted in no effect on ADG or G:F, even though feed intake was increased compared with pigs fed negative control diets [55].

The variation in the growth performance results reported in previous studies may be explained by differences in the concentration or chemical form of other bioactive molecules present in these feed additives. In our study, both SOE and BOE were included in the diet at a concentration less than 0.04%, which is less than doses of 0.1% to 0.3% that resulted in growth improvements in other experiments [7,55].

In EXP 2, we evaluated milk-derived substances fed to nursery pigs. Unfortunately, the methods used for the chemometrics analysis did not result in the identification of any unique compounds in these products, making it difficult to determine if they contained any potentially beneficial compounds that may promote growth in pigs.

The chemometric analysis of the yeast-based products showed that they contained substantial concentrations of free amino acids and dipeptides, but no unique chemical compounds. However, the methods we used for characterizing the compounds present in yeast were not appropriate for measuring nucleotides, mannan oligosaccharides, and β-glucans, which are known to be present in yeast cell walls [2].

### 4.2. Growth Performance Responses

There were no differences in growth performance between any of the non-antibiotic dietary treatments and the NC for each experiment. The lack of growth performance improvements from feeding these additives may have been due to the high health of pigs and hygiene conditions of facilities used during these experiments. Several studies have shown that growth improvements from feeding diets containing antibiotic growth promoters are greater on commercial farms where pigs are often challenged with disease insults than in university or commercial research settings [56]. The lack of growth performance responses could therefore be attributed to cleaner facilities, less stress, and a lower disease pressure on research farms compared with the conditions on commercial farms [56]. When considering the absence of mortality observed in these EXP, it is possible that the biologically active components of these additives provided no benefits under near the optimal health, environment, and nutrition conditions provided. However, these results may not be surprising. The results from a review and summary of more than 2000 published swine feeding trials showed that only 28% of the trials reported improvements in ADG, 67% of trials showed no change, and 3% reported reductions in growth from adding several different types of feed additives to swine diets as alternatives to growth-promoting levels of antibiotics [57].

### 4.3. Serum Metabolite Responses

Greater concentrations of lysine were only observed in serum of pigs fed PC compared with those fed other dietary treatments in EXP 3. The results from previous studies have suggested that a change in serum lysine concentration can be directly related to a change in dietary intake of lysine [58], which may have occurred because of the increased feed intake observed in EXP 3. In that study, feeding diets containing antibiotics increased ADFI by 6.7%, which corresponded to a 25% increase in serum lysine concentration. However, serum lysine concentration is also affected by a slower rate of catabolism compared with that of other essential amino acids [58]. Therefore, the slower rate of lysine catabolism could explain why the magnitude of change in serum lysine concentration was greater than the increase attributed to increased feed intake.

In addition to serum lysine concentrations, the only other metabolites identified as biomarkers for each treatment were antibiotic metabolites from feeding PC diets. Despite the potential for feeding diets containing PHY01, PHY02, PHY03, TUM, GAR, SOE, and BOE to reduce oxidative stress, no difference in serum metabolites and enzymes, such as protein carbonyls, malondialdehyde, superoxide dismutase, or glutathione peroxidase, were observed compared with feeding NC, which indicated that pigs were not experiencing oxidative stress in any of the EXP [59,60].

### 4.4. Cecal Metabolite and Microbiome Responses

Ibervirin was the only metabolite identified in both the chemometric analysis and as a biomarker for the GAR treatment in cecal contents but not in serum, suggesting that ibervirin was not absorbed in the small intestine. Ibervirin is a thiocyanate deemed to be a biomarker of root vegetable consumption in humans, but there is no information available on its impact on animal health and performance [61].

None of the other metabolites identified in the chemometrics analysis of the feed additives were subsequently identified as biomarkers for any of the other dietary treatments. For example, curcumin was identified as being predominantly present in TUM, but it did not have a significant indicator value in the cecal contents of the pigs fed this product. Curcumin is poorly digested, and if it is included in the diet, it should be present in the intestinal contents and could potentially alter the gut microbiome, as shown previously [62]. It is possible that if TUM had been included at a higher concentration in the diet, curcumin may have had a measurable effect on the cecal microbiome. This expected effect has been shown in previous experiments when feeding a greater dose of curcumin (0.03%) compared with the 0.01% concentration of turmeric added to the diets in our study [63]. Although we followed the product manufacturers’ recommended dietary inclusion rates for all feed additives evaluated, the concentrations of each active compound provided to the diets may have been too low to result in detectable concentrations of flavonoids in the cecal contents of pigs fed the BOE and SOE diets or the lack of quinic acid in the cecal contents of pigs fed the PHY06 and VSM02 additives. The lack of these potentially biologically active compounds in intestinal contents may also explain the minimal impact observed on the microbiome.

There were no differences in the weighted Bray–Curtis beta-diversity metrics for cecal (EXP 1 *p* = 0.50; EXP 2 *p* = 0.38; EXP 3 *p* = 0.15) or ileal (EXP 1 *p* = 0.42; EXP 2 *p* = 0.33; EXP 3 *p* = 0.86) content samples in any of the three EXP conducted (Figure 1 and Figure 3). There were also no differences in the rarefied richness (cecal: EXP 1 *p* = 0.98, EXP 2 *p* = 0.11, EXP 3 *p* = 0.82; ileal: EXP 1 *p* = 0.39, EXP 2 *p* = 0.23, EXP 3 *p* = 0.52) or Simpson diversity index (cecal: EXP 1 *p* = 0.37, EXP 2 *p* = 0.78, EXP 3 *p* = 0.68; ileal: EXP 1 *p* = 0.81, EXP 2 *p* = 0.46, EXP 3 *p* = 0.87) between dietary treatments in either ileal or cecal content samples for any of the three EXP (Figure 2 and Figure 4). The reasons behind the lack of significant microbiome responses are not clear, but in addition to relatively low dietary doses of the bioactive compounds previously discussed, other technical aspects of the EXP may be considered. For example, barriers between pens consisted of bars with open spaces instead of solid barriers between pens of pigs fed different treatments. In poultry, alpha diversity of the cecal microbiome was not different between birds fed negative control diets compared with those fed positive control diets containing medium-chain fatty acids when the EXP was conducted in facilities that separated pens with mesh screens [64]. However, when the same EXP was conducted in facilities with solid barriers between pens, or in isolation facilities, the alpha diversity of the cecal microbiome was significantly decreased by feeding the diet containing medium-chain fatty acids [60]. These findings suggest the type of pen divisions used in the facilities where these EXP were conducted may have limited our ability to detect differences in microbiome composition due to dietary treatment because pigs had the opportunity to be in contact with pigs in adjacent pens fed other dietary treatments, which provided opportunities to share microbes.

Despite a lack of differences in alpha and beta diversity, some dietary feed additives affected specific taxa of bacteria, indicating that specific dietary treatments had selective effects on the bacterial communities. For example, feeding antibiotic growth promoters to pigs has been shown to increase the abundance of *Proteobacteria*, and particularly *Escherichia* coli, in the gastrointestinal tract [65]. The results from a similar research trial showed that feeding a combination of chlortetracycline, sulfamethazine, and penicillin antibiotics to pigs resulted in selection of *Lachnobacterium* in the ileum, compared with pigs fed positive and negative control diets [66]. Our results indicate that including feed additives in the diets of high-health weaned pigs did not greatly alter the cecal and ileal microbiome, with minimal effects on overall microbial composition, including pigs fed diets containing antibiotics. However, selective effects on specific bacterial taxa were observed, which is consistent with the results from previous studies.

For the cecal content samples in EXP 3, bacteria from the family *Streptomycetaceae* distinguished pigs fed PC diets from those fed all other treatments. This family of bacteria is commonly found in soil, and it is also used to produce commercial antibiotics [67]. In addition, there were significant increases (*p* < 0.05) in the abundance of ASVs affiliated with the genus *Lactobacillus* when feeding PHY01, PHY03, and GAR in EXP 1. *Lactobacillus* spp. have been extensively studied and evaluated as dietary probiotic additives. Multiple studies have reported that when feeding various strains of *Lactobacillus* spp. to nursery pigs, the growth rate and feed efficiency were improved, diarrhea was reduced, and resistance to gastrointestinal pathogens was increased [68]. In our study, one ASV of *Lactobacillus* showed increased relative abundance when the phytogenic products were added to diets, which was a result not observed in pigs fed PC. Although this increase was not associated with improved growth performance in this EXP, it is possible that in a disease challenge situation, the increase in *Lactobacillus* may provide a beneficial effect. Adding various *Lactobacillus* strains to swine and poultry diets has been shown to have an antagonistic effect on pathogenic bacteria in the intestine, including *Salmonella,* which may be due to its reported antimicrobial effects [16,69,70]. This observation indicates that given the potential protective properties of the *Lactobacillus* strains, pigs may benefit from diets containing PHY01, PHY03, or GAR when experiencing a pathogen challenge.

Relative abundance of *Megasphaera* and *Prevotella stercorea* increased (*p* < 0.05) in pigs fed YN01 compared with those fed NC. The mannan content in yeast cell walls likely contributed to the increased abundance of these two taxa because they are associated with fermentative roles [71].

## 5. Conclusions

Of the 13 feed additives evaluated, none of them significantly improved the growth performance of nursery pigs under the high health and environmental hygiene conditions used in this study, compared with the dietary addition of conventional growth-promoting antibiotics. In addition, these feed additives had a minimal impact on the serum and cecal metabolome profiles and had no significant impact on bacterial community composition in ileal or cecal contents. Despite a lack of responses on the microbiome, supplementing the herbal blends PHY01, PHY03, and the phytogenic extract GAR increased the relative abundance of bacteria in the genus *Lactobacillus*. It is unknown if the responses would be different if these feed additives were fed to weaned pigs undergoing health challenge.

## Figures and Tables

**Figure 1 animals-14-00060-f001:**
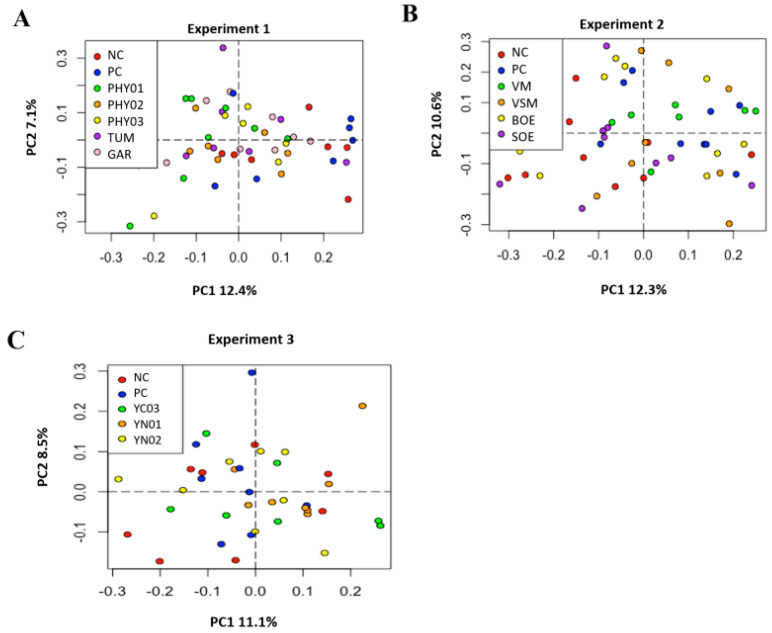
Principal component analyses (weighted Bray–Curtis distances) of bacterial community composition in cecal samples of pigs fed different dietary additives. (**A**–**C**) represent analyses from Experiments 1–3. Key in A shows antibiotics (PC), no antibiotics (NC), essential oil and herb mixture 1 (PHY01), essential oil and herb mixture 2 (PHY02), essential oil and herb mixture 3 (PHY03), turmeric (TUM), and garlic (GAR). Lack of separation indicates no differences in microbiome composition among treatments (PERMANOVA, R^2^ = 0.127, *p* = 0.50). Key in B shows antibiotics (PC), no antibiotics (NC), bitter orange extract (BOE), sweet orange extract (SOE), volatile milk substances (VM), and volatile and semi-volatile milk substances (VSM). PERMANOVA, R^2^ = 0.098, *p* = 0.383. Key in C shows antibiotics (PC), no antibiotics (NC), yeast nucleotide product 1 (YN01), yeast nucleotide product 2 (YN02), and yeast cell wall (YC03) product. PERMANOVA, R^2^ = 0.118, *p* = 0.147.

**Figure 2 animals-14-00060-f002:**
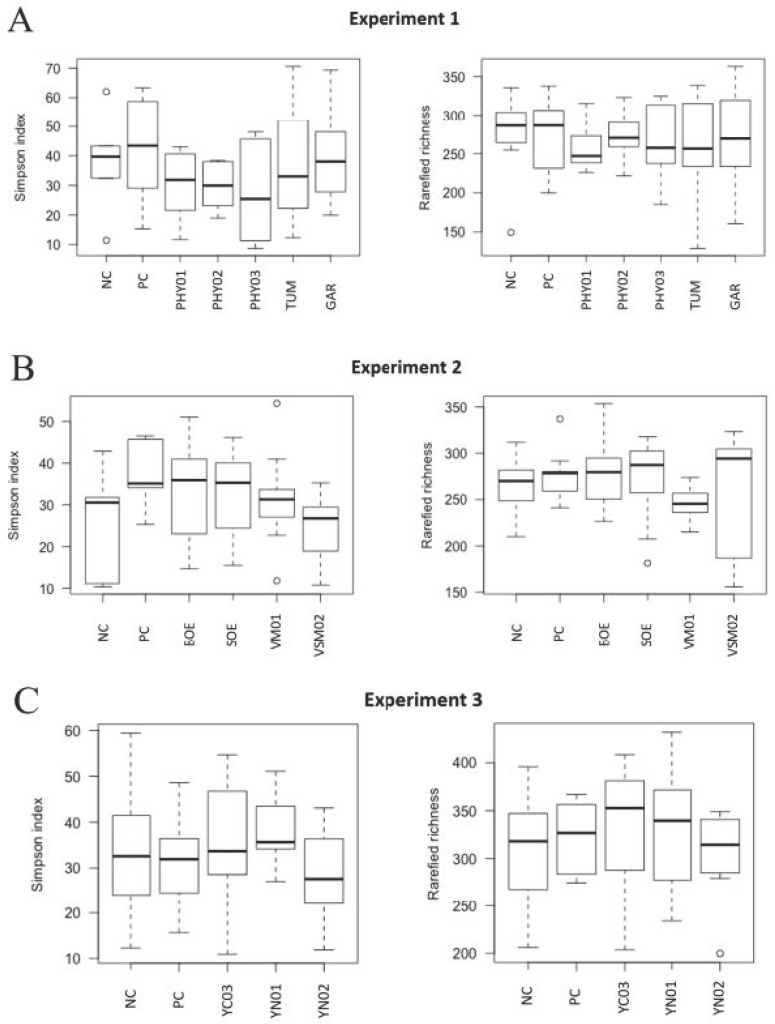
**Alpha diversity (rarefied richness and Simpson index) in cecal samples of pigs fed different dietary additives**. (**A**–**C**) represent analyses from Experiments 1–3. Treatments in A include antibiotics (PC), no antibiotics (NC), essential oil and herbal mixture 1 (PHY01), essential oil and herbal mixture 2 (PHY02), essential oil and herbal mixture 3 (PHY03), turmeric (TUM), garlic (GAR). Treatments in B include antibiotics (PC), no antibiotics (NC), bitter orange extract (BOE), sweet orange extract (SOE), volatile milk-derived substances (VM01), and volatile and semi-volatile milk-derived substances (VSM02). Treatments in C include antibiotics (PC), no antibiotics (NC), yeast nucleotide product 1 (YN01), yeast nucleotide product 2 (YN02), and yeast cell wall (YC03) product.

**Figure 3 animals-14-00060-f003:**
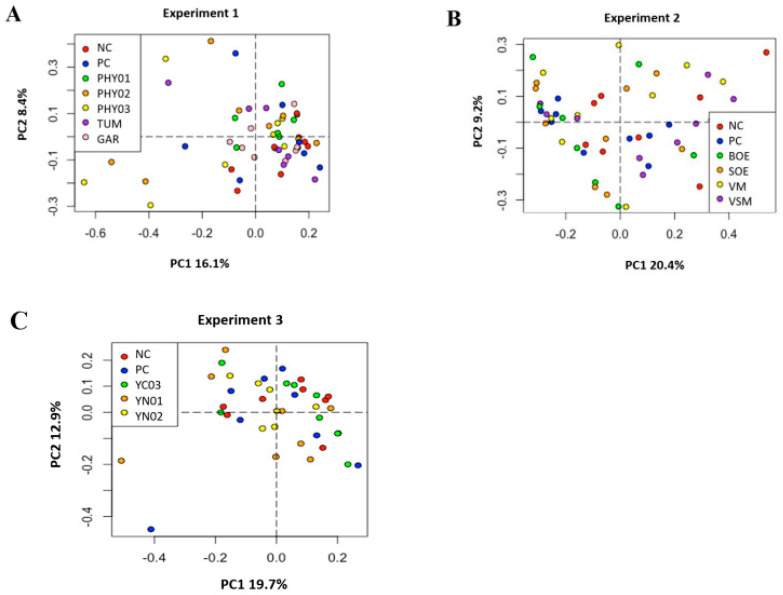
Principal component analyses (weighted Bray–Curtis distances) of bacterial community composition in ileal samples of pigs fed different dietary additives. (**A**–**C**) represent analyses from Experiment 1–3. Key in A shows antibiotics (PC), no antibiotics (NC), essential oil and herbal mixture 1 (PHY01), essential oil and herbal mixture 2 (PHY02), essential oil and herbal mixture 3 (PHY03), turmeric (TUM), and garlic (GAR). Lack of separation indicates no differences in microbiome composition among treatments (PERMANOVA, R^2^ = 0.112, *p* = 0.417). Key in B shows antibiotics (PC), no antibiotics (NC), bitter orange extract (BOE), sweet orange extract (SOE), volatile milk-derived substances (VM), and volatile and semi-volatile milk-derived substances (VSM). PERMANOVA, R^2^ = 0.101, *p* = 0.334. Key in C shows antibiotics (PC), no antibiotics (NC), yeast nucleotide product 1 (YN01), yeast nucleotide product 2 (YN02), and yeast cell wall (YC03) product. PERMANOVA, R^2^ = 0.078, *p* = 0.855.

**Figure 4 animals-14-00060-f004:**
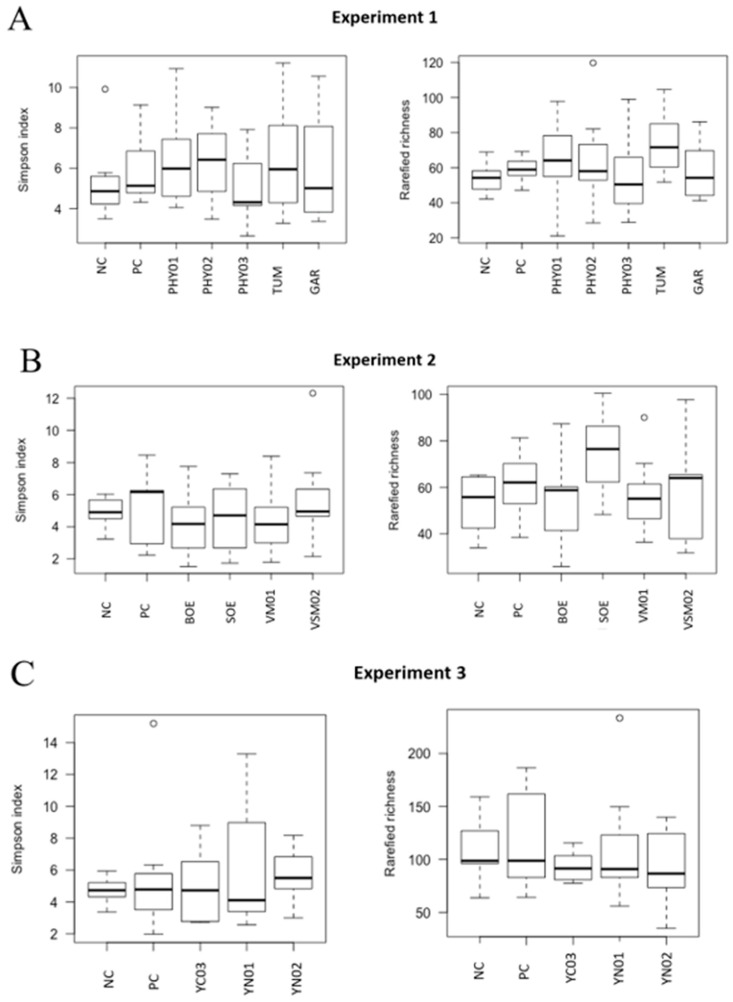
**Alpha diversity (rarefied richness and Simpson index) in ileal samples of pigs fed different dietary additives**. (**A**–**C**) represent analyses from Experiment 1–3. Treatments in A include antibiotics (PC), no antibiotics (NC), essential oil and herbal mixture 1 (PHY01), essential oil and herbal mixture 2 (PHY02), essential oil and herbal mixture 3 (PHY03), turmeric (TUM), and garlic (GAR). Treatments in B include antibiotics (PC), no antibiotics (NC), bitter orange extract (BOE), sweet orange extract (SOE), volatile milk-derived substances (VM01), and volatile and semi-volatile milk-derived substances (VSM02). Treatments in C include antibiotics (PC), no antibiotics (NC), yeast nucleotide product 1 (YN01), yeast nucleotide product 2 (YN02), and yeast cell wall (YC03) product.

**Table 1 animals-14-00060-t001:** Summary of dietary treatments.

Feed Additive	Treatment Abbreviations	Experiment	Treatment Description	Diet Inclusion Rate
Controls	Positive control (PC)	1, 2, 3	Antibiotic (0.01% chlortetracycline and 0.1% sulfamethazine	0.5%
	Negative control (NC)	1, 2, 3	No antibiotics or feed additives	N/A
Herbal blends	PHY01	1	NC + Essential oil and herbal blend 1	0.03%
	PHY02	1	NC + Essential oil and herbal blend 2	0.1%
	PHY03	1	NC + Essential oil and herbal blend 3	0.02%
Phytogenic extracts	TUM	1	NC + Turmeric	0.01%
	GAR	1	NC + Garlic	0.015%
	BOE	2	NC + Bitter orange extract	0.03%
	SOE	2	NC + Sweet orange extract	0.018 to 0.037%
Milk-derived substances	VM01	2	NC + Volatile milk-derived substances	0.03 to 0.05%
	VSM02	2	NC + Volatile and semi-volatile milk-derived substances	0.03 to 0.05%
Yeast products	YN01	3	NC + Yeast nucleotide 1	0.05 to 0.1%
	YN02	3	NC + Yeast nucleotide 2	0.05 to 0.1%
	YC03	3	NC + Yeast cell walls	0.05 to 0.1%

**Table 2 animals-14-00060-t002:** Body weight (BW), average daily gain (ADG), average daily feed intake (ADFI), and gain:feed (G:F) of nursery pigs fed diets with essential oils and herbal blends and phytogenic extracts.

		Experiment 1 Dietary Treatments *								*p*-Value		
Measure	Days	Positive Control	Negative Control	PHY01	PHY02	PHY03	TUM	GAR	SEM	Treatment	Time	Treatment × Time
BW, kg	0	6.69	6.69	6.69	6.69	6.69	6.68	6.70				
	10	8.26	8.48	8.58	8.40	8.36	8.50	8.53	1.33	0.641	<0.001	0.565
	21	13.61	13.89	13.97	13.63	13.66	14.06	14.10				
	42	27.29	26.86	26.80	26.56	26.75	27.29	27.66				
ADG, kg	0 to 10	0.16	0.18	0.19	0.17	0.17	0.18	0.18				
	10 to 21	0.49	0.49	0.49	0.47	0.48	0.50	0.50	0.02	0.489	<0.001	0.617
	21 to 42	0.65	0.62	0.61	0.62	0.62	0.63	0.65				
ADFI, kg	0 to 10	0.18	0.19	0.20	0.19	0.18	0.19	0.20				
	10 to 21	0.54	0.56	0.58	0.55	0.55	0.57	0.58	0.04	0.316	<0.001	0.712
	21 to 42	0.90	0.89	0.89	0.90	0.91	0.92	0.93				
G:F	0 to 10	0.88 ^b^	0.94 ^a^	0.93 ^a^	0.91 ^ab^	0.91 ^ab^	0.93 ^a^	0.95 ^a^				
	10 to 21	0.91 ^a^	0.88 ^ab^	0.84 ^b^	0.88 ^ab^	0.89 ^ab^	0.89 ^ab^	0.87 ^ab^	0.01	0.828	<0.001	0.049
	21 to 42	0.72	0.72	0.69	0.69	0.69	0.69	0.69				

* Dietary treatments include antibiotics (positive control), no antibiotics (negative control), essential oils and herbal blend 1 (PHY01), essential oils and herbal blend 2 (PHY02), essential oils and herbal blend 3 (PHY03), turmeric (TUM), and garlic (GAR). ^a,b^ Means within a row with uncommon superscripts differ (*p* < 0.05).

**Table 3 animals-14-00060-t003:** Body weight (BW), average daily gain (ADG), average daily feed intake (ADFI), and gain:feed (G:F) of nursery pigs fed diets with phytogenic extracts and milk-derived substances.

		Experiment 2 Dietary Treatments *							*p*-Value		
Measure	Days	Positive Control	Negative Control	BOE	SOE	VM01	VSM02	SEM	Treatment	Time	Treatment × Time
BW, kg	0	6.60	6.60	6.61	6.61	6.60	6.61				
	10	7.98	8.10	7.96	7.82	7.97	7.92	1.63	0.038	<0.001	0.161
	21	13.06 ^a^	12.78 ^ab^	12.54 ^ab^	12.15 ^b^	12.66 ^ab^	13.06 ^ab^				
	42	27.26 ^a^	26.22 ^b^	26.11 ^b^	24.97 ^c^	26.54 ^ab^	25.83 ^b^				
ADG, kg	0 to 10	0.14	0.15	0.14	0.12	0.14	0.13				
	10 to 21	0.46 ^a^	0.42 ^b^	0.42 ^b^	0.40 ^b^	0.43 ^ab^	0.42 ^b^	0.04	0.001	<0.001	0.625
	21 to 42	0.68 ^a^	0.64 ^bc^	0.65 ^ab^	0.61 ^c^	0.66 ^ab^	0.63 ^bc^				
ADFI, kg	0 to 10	0.17	0.17	0.17	0.16	0.16	0.17				
	10 to 21	0.51 ^a^	0.49 ^ab^	0.47 ^ab^	0.45 ^b^	0.49 ^ab^	0.46 ^b^	0.05	0.035	<0.001	0.456
	21 to 42	0.95 ^a^	0.90 ^bc^	0.90 ^ab^	0.85 ^c^	0.92 ^abc^	0.90 ^bc^				
G:F	0 to 10	0.83	0.87	0.81	0.77	0.83	0.78				
	10 to 21	0.90	0.87	0.89	0.86	0.87	0.90	0.02	0.085	<0.001	0.318
	21 to 42	0.72	0.72	0.72	0.72	0.72	0.71				

* Dietary treatments include antibiotics (positive control), no antibiotics (negative control), bitter orange extract (BOE), sweet orange extract (SOE), volatile milk-derived substances (VM01), and volatile and semi-volatile milk-derived substances (VSM02). ^a,b,c^ Means within a row with uncommon superscripts differ (*p* < 0.05).

**Table 4 animals-14-00060-t004:** Body weight (BW), average daily gain (ADG), average daily feed intake (ADFI), and gain:feed (G:F) of nursery pigs fed diets with yeast products.

		Experiment 3 Dietary Treatments *						*p*-Value		
Measure	Days	Positive Control	Negative Control	YN01	YN02	YC03	SEM	Treatment	Time	Treatment × Time
BW, kg	0	6.21	6.19	6.20	6.20	6.20				
	10	7.60	7.48	7.53	7.41	7.50	1.79	0.029	<0.001	0.041
	21	12.61	12.08	12.08	12.18	12.00				
	42	26.40 ^a^	24.24 ^b^	24.56 ^b^	24.15 ^b^	24.20 ^b^				
ADG, kg	0 to 10	0.14	0.13	0.13	0.12	0.13				
	10 to 21	0.45 ^a^	0.42 ^ab^	0.41 ^ab^	0.43 ^ab^	0.41 ^b^	0.05	0.001	<0.001	0.460
	21 to 42	0.65 ^a^	0.58 ^b^	0.60 ^b^	0.57 ^b^	0.58 ^b^				
ADFI, kg	0 to 10	0.17	0.16	0.16	0.15	0.16				
	10 to 21	0.54	0.48	0.50	0.50	0.49	0.06	0.017	<0.001	0.219
	21 to 42	1.00 ^a^	0.87 ^b^	0.88 ^b^	0.90 ^b^	0.89 ^b^				
G:F	0 to 10	0.84 ^a^	0.80 ^ab^	0.82 ^ab^	0.77 ^b^	0.79 ^ab^				
	10 to 21	0.85	0.86	0.83	0.86	0.83	0.02	0.054	<0.001	0.640
	21 to 42	0.66	0.67	0.68	0.63	0.65				

* Dietary treatments include antibiotics (positive control), no antibiotics (negative control), yeast nucleotide product 1 (YN01), yeast nucleotide 2 (YN02), and yeast cell wall (YC03) product. ^a,b^ Means within a row with uncommon superscripts differ (*p* < 0.05).

**Table 5 animals-14-00060-t005:** Cecal bacterial strains with high abundance and specificity to pigs fed antibiotics, essential oils and herbal blends, phytogenic extracts, milk-derived substances, and yeast products compared with pigs fed negative control diets among experiments.

Bacterial Strain	Indicator Value ^1^	Experiment	Treatment ^2^	Negative Control Mean ^3^ ± Standard Deviation	Treatment Mean ^3^ ± Standard Deviation	*p* Value ^4^
*Lactobacillus* spp.	0.878	1	PHY01	0.755 ± 0.614	5.449 ± 4.621	0.019
*Lactobacillus* spp.	0.940	1	PHY01	0.755 ± 0.614	11.925 ± 10.770	0.010
*Lactobacillus* spp.	0.835	1	GAR	0.755 ± 0.614	3.835 ± 3.681	0.046
*Ruminococcus* spp.	0.821	1	GAR	0.094 ± 0.16	0.434 ± 0.342	0.032
*SMB53* spp.	0.820	2	PC	0.495 ± 0.474	2.26 ± 1.606	0.009
*Streptomycetaceae*	1.000	3	PC	0 ± 0	0.033 ± 0.019	0.001
*Megasphaera* spp.	0.915	3	YN01	0.228 ± 0.321	2.438 ± 2.175	0.019
*Prevotella* *stercorea*	0.817	3	YN01	0.035 ± 0.045	0.157 ± 0.12	0.024

^1^ The indicator value is a product of frequency and abundance and ranges from 0 to 1. Greater indicator values show that a given taxon is more abundant in all samples from a given group compared with other groups. The table shows indicator values greater than 0.8, indicating that the selected taxa are faithful biomarkers of a given treatment. ^2^ Dietary treatments include antibiotics (PC), essential oil and herbal mixture 1 (PHY01), essential oil and herbal mixture 3 (PHY03), garlic (GAR), and yeast nucleotide product 1 (YN01). ^3^ Means expressed as relative abundance (%). ^4^ *p*-value from Kruskal–Wallis one-way analysis of variance.

## Data Availability

A summary of chemical compounds identified in chemometrics analysis of feed additives, along with metabolites with high abundance and specificity identified in serum and cecal contents of pigs fed various feed additives, are provided in the Appendix A. The 16S rRNA sequence data analyzed in this manuscript are available in NCBI Sequence Read Archive under accession BioProject ID PRJNA950003. The link is: https://dataview.ncbi.nlm.nih.gov/object/PRJNA950003?reviewer=qc9rohdoa3u4huurtn063r90il (accessed on 29 March 2023).

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
