# Peer review of "Growth Performance, Metabolomics, and Microbiome Responses of Weaned Pigs Fed Diets Containing Growth-Promoting Antibiotics and Various Feed Additives"

_animals, 2023, doi:10.3390/ani14010060_

Round 1

Reviewer 1 Report (Previous Reviewer 1)

Comments and Suggestions for Authors

I would like to congratulate the authors on resubmitting this interesting paper. As the minor corrections I pointed have been addressed I am recomending the paper for publication. I aggree with the authors that "lack" of significant differences between treatments is also a result and should be of public knowledge. 

Author Response

Thank you!

Reviewer 2 Report (Previous Reviewer 4)

Comments and Suggestions for Authors

None identified

Author Response

Thank you!

Reviewer 3 Report (New Reviewer)

Comments and Suggestions for Authors

Growth performance, metabolomics, and microbiome responses of weaned pigs fed diets containing growth promoting antibiotics and various feed additives

Dear Authors,

The manuscript is interesting and describes comparison between AGP (positive control) and feed additives which in EU replaced it from 1st  Jan 2006 and 1st Jan 2017 in US . The DOE is prepared properly, but main problem for me is the statistical analysis (ANOVA) in case of growth performance of nursery pigs (BW, ADG, ADFI, G:F). Especially in the first experiment conducted with 7 treatments and 8 replications in each, significant difference between AGP (positive control) and other six treatments wasn’t confirmed. Additionally p-value is determined only for one value between ie. BW at 0, 10, 21 and 42nd day of experiment. It must be determined, i.e for each day of measurement.

The Introduction, The Materials and Methods are quite good prepared, some changes are needed to add in subsection 2.4. Statistical analysis of growth performance data.

Below I add some suggestions:

Line 24-25

In text is sentence: “…None of the feed additives except antibiotics improved growth performance compared with feeding NC…”, but in table 1 values BW are better for GAR than in positive control and in day 42 for TUM equal with the positive control, earlier TUM brings better results (statistically non-significant, but with higher number of replications and lower number of treatments that could be to prove from statistical point of view, probably it will be worth to repeat that first experiment to additionally relate those results with increase number of Lactobacillus sp. described in the Conclusions).

Line 59-65

The reference no.4 appears three times in one paragraph, maybe is possible to find other one or two  references instead of  Windisch et al. 2008.

More references from last 3-4 years are necessarily to added.

Line 167

Table 1

Maybe better to separate three values in the last column (pens per treatment) for each experiment, because animals in each have different variability and it is not possible to compare them in one analysis or it is possible also to delete this column and left only five first columns.

Line 185-189

The DOE is prepared properly, when checking number of replications in treatments. Information about Two-way ANOVA needed, but in my opinion better will be to use in analysis one-way ANOVA because time factor is obvious. During 42 days of growth difference in BW between 0 and 42th day of experiment must be very large (respectively ± 6,5 kg and ± 27,0 kg). In case of ANOVA (normal distribution determined, homogeneity of variance?) p-value must be lower in some cases even than 0,001 (i.e. 0,00000034,…, 0,000 could be also acceptable).

Line 303

More accurate analysis needed, the p-value for each day of measurement, and all rows in case ADG, ADFI, G:F (FCR). Without time, because describing significance level in table by post-hoc test is not possible to present. It is possible to using in tables with two factors, but in this case you need add mean values separately for additives and separately mean values for time of measurement (but in case of time it is obvious and in 3 experiments only one interaction was found), but if  You use time as an independent variable in analysis You don’t have dependent (you need to have 2 factors and estimator calculated on basis those two factors. Concluded: one-way ANOVA, and changes in time could present in form of line charts for all treatments.

Possibly The Results could be needed to modified or even The Discussion.

Line 310

More accurate analysis needed, p-value for each day of measurement, and all row in case ADG, ADFI, G:F (FCR) like in line 303.

Line 316

More accurate analysis needed, p-value for each day of measurement, and all row in case ADG, ADFI, G:F (FCR) like in line 303.

Line 443

Please centre text horizontally and vertically (Layout, Alignment) in cells of table.

Line 583-585

In The Conclusions is sentence: ‘…Of the 13 feed additives evaluated, none of them improved growth performance of nursery pigs under the high health and environmental hygiene conditions used in this study, compared with the dietary addition of conventional growth promoting antibiotics…’

In first experiment no significant differences was found, but garlic extract treatment nursery pigs have higher BW in comparison to positive control, almost the same situation with Turmeric: higher (non-significant) BW at day 0, 10 and 21, the same value with positive control in day 42. Maybe is possible in future to repeat first experiment with 4 treatments: positive control, negative control, TUM and GAR or mentioned below PHY01 and PHY03 (56 pens/4 = 14 replications in each treatment, 9-12 nursery pigs in each replication as is mentioned in last sentences to check if: “…supplementing the herbal blends PHY01, PHY03, and the phytogenic extract GAR increased the relative abundance of bacteria in the genus Lactobacillus. It is unknown if the responses would be different if these feed additives were fed to weaned pigs undergoing a health challenge”?

Author Response

Dear Authors,

The manuscript is interesting and describes comparison between AGP (positive control) and feed additives which in EU replaced it from 1st  Jan 2006 and 1st Jan 2017 in US . The DOE is prepared properly, but main problem for me is the statistical analysis (ANOVA) in case of growth performance of nursery pigs (BW, ADG, ADFI, G:F). Especially in the first experiment conducted with 7 treatments and 8 replications in each, significant difference between AGP (positive control) and other six treatments wasn’t confirmed. Additionally p-value is determined only for one value between ie. BW at 0, 10, 21 and 42nd day of experiment. It must be determined, i.e for each day of measurement.

Response: We followed standard statistical power test procedures to determine the minimum number of replications per treatment in each experiment as described in section 2.4. We also analyzed the growth performance data correctly as a randomized complete block design using the GLIMMIX procedure of SAS with repeated measures in time where replicate was a random effect and treatment were fixed effects in the model. Your comment suggests that you don’t understand the statistical analysis model that we used. No changes.

The Introduction, The Materials and Methods are quite good prepared, some changes are needed to add in subsection 2.4. Statistical analysis of growth performance data.

Response: See previous comment. No changes.

Below I add some suggestions:

Line 24-25 In text is sentence: “…None of the feed additives except antibiotics improved growth performance compared with feeding NC…”, but in table 1 values BW are better for GAR than in positive control and in day 42 for TUM equal with the positive control, earlier TUM brings better results (statistically non-significant, but with higher number of replications and lower number of treatments that could be to prove from statistical point of view, probably it will be worth to repeat that first experiment to additionally relate those results with increase number of Lactobacillus sp. described in the Conclusions).

Response: We do not acknowledge or discuss numerical differences among treatments. As previously indicated, we used an appropriate statistical power test to determine the minimum number of replicates needed per treatment in each experiment to detect significant differences if they existed. We added the word “significant” on line 24 to clarify. Perhaps conducting this experiment again with more replications may result in a significant difference between the treatments that you reference, but this is only speculation.

Line 59-65 The reference no.4 appears three times in one paragraph, maybe is possible to find other one or two  references instead of  Windisch et al. 2008. More references from last 3-4 years are necessarily to added.

Response: You are correct. Reference 4 is a review paper that extensively summarized results from published studies involving feeding diets containing phytogenic compounds and is a key resource for readers to learn more about previous studies on this topic. Therefore, we believe it is worth referencing more than once. Furthermore, several other references in addition to reference 4 were included in this paragraph. No changes made.

Line 167 Table 1-Maybe better to separate three values in the last column (pens per treatment) for each experiment, because animals in each have different variability and it is not possible to compare them in one analysis or it is possible also to delete this column and left only five first columns.

Response: We agree. This information was described in the methods and the last column of this table was deleted. 

Line 185-189 The DOE is prepared properly, when checking number of replications in treatments. Information about Two-way ANOVA needed, but in my opinion better will be to use in analysis one-way ANOVA because time factor is obvious. During 42 days of growth difference in BW between 0 and 42th day of experiment must be very large (respectively ± 6,5 kg and ± 27,0 kg). In case of ANOVA (normal distribution determined, homogeneity of variance?) p-value must be lower in some cases even than 0,001 (i.e. 0,00000034,…, 0,000 could be also acceptable).

Response: We analyzed and reported these data correctly. The P value for time was rounded to 0.001, but to clarify, the P value was actually lower than that in the time section of the table. Therefore, a < sign was added to the Time column of P values in tables 2, 3, and 4.

Line 303 More accurate analysis needed, the p-value for each day of measurement, and all rows in case ADG, ADFI, G:F (FCR). Without time, because describing significance level in table by post-hoc test is not possible to present. It is possible to using in tables with two factors, but in this case you need add mean values separately for additives and separately mean values for time of measurement (but in case of time it is obvious and in 3 experiments only one interaction was found), but if  You use time as an independent variable in analysis You don’t have dependent (you need to have 2 factors and estimator calculated on basis those two factors. Concluded: one-way ANOVA, and changes in time could present in form of line charts for all treatments. Possibly The Results could be needed to modified or even The Discussion.

Response: Your comments indicate that you don’t understand the importance of using repeated measures over time (see Littell et al. (1998) Statistical analysis of repeated measures data using SAS procedures doi:10.2527/1998.7641216x). Time was used in the model as a repeated measure which is why we did not present the overall average (combining all treatments) by time. The p value for time was included in this way because it was a significant variable in the model. Because the analysis was done using repeated measures, p value represents the overall experiment. No changes.

Line 310 More accurate analysis needed, p-value for each day of measurement, and all row in case ADG, ADFI, G:F (FCR) like in line 303.

Response: See previous response.

Line 316 More accurate analysis needed, p-value for each day of measurement, and all row in case ADG, ADFI, G:F (FCR) like in line 303.

Response: See previous response on analysis using repeated measures.

Line 443 Please centre text horizontally and vertically (Layout, Alignment) in cells of table.

Response: Text in all tables was centered.

Line 583-585 In The Conclusions is sentence: ‘…Of the 13 feed additives evaluated, none of them improved growth performance of nursery pigs under the high health and environmental hygiene conditions used in this study, compared with the dietary addition of conventional growth promoting antibiotics…’In first experiment no significant differences was found, but garlic extract treatment nursery pigs have higher BW in comparison to positive control, almost the same situation with Turmeric: higher (non-significant) BW at day 0, 10 and 21, the same value with positive control in day 42. Maybe is possible in future to repeat first experiment with 4 treatments: positive control, negative control, TUM and GAR or mentioned below PHY01 and PHY03 (56 pens/4 = 14 replications in each treatment, 9-12 nursery pigs in each replication as is mentioned in last sentences to check if: “…supplementing the herbal blends PHY01, PHY03, and the phytogenic extract GAR increased the relative abundance of bacteria in the genus Lactobacillus. It is unknown if the responses would be different if these feed additives were fed to weaned pigs undergoing a health challenge”?

Response: As previously indicated, we do not acknowledge or discuss numerical differences among treatments. This is why statistical analysis is used in experiments. We used an appropriate statistical power test to determine the minimum number of replicates needed per treatment in each experiment to detect significant differences if they existed. We added the word “significant” on line 609 to clarify. Perhaps conducting this experiment again with more replications may result in a significant difference between the treatments that you reference, but this is only speculation.

Reviewer 4 Report (New Reviewer)

Comments and Suggestions for Authors

There were no differences in growth performance between any of the non-antibiotic dietary treatments and the NC for each experiment, so what is the significance of measuring metabolomics and microbiome responses?

Comments on the Quality of English Language

 Moderate editing of English language required

Author Response

We were expecting to find some improvements in growth performance prior to conducting these experiments which led us to collecting samples and assessing potential changes in the metabolome and microbiome. Although we observed minimal growth performance improvements, we believe it is still worthwhile to conduct a metabolome and microbiome analysis to corroborate that feeding these compounds had minimal effects on the microbiome and metabolome which explains the lack of growth responses. 

Round 2

Reviewer 3 Report (New Reviewer)

Comments and Suggestions for Authors

The statistical calculations do not include all results from Table 2. Table 2 shows only the results from day 10. Where are the statistical analysis results for 21st and 42nd day? In this case, the data from the three periods cannot be combined and presented as one probability value (unless it is a trade secret, in which case it should be added as information, but I doubt a serious journal would accept the lack of comparisons of this type). Values should be presented separately for all days (eventually periods): 10th, 21st, 42nd. Information about the significance of differences and homogeneity of variance on day 0, as well as about the transmission of AGPs into the environment and their possible entry into the human body by microorganisms will be also valuable information in future.

Author Response

The statistical calculations do not include all results from Table 2. Table 2 shows only the results from day 10. Where are the statistical analysis results for 21st and 42nd day? In this case, the data from the three periods cannot be combined and presented as one probability value (unless it is a trade secret, in which case it should be added as information, but I doubt a serious journal would accept the lack of comparisons of this type). Values should be presented separately for all days (eventually periods): 10th, 21st, 42nd. Information about the significance of differences and homogeneity of variance on day 0, as well as about the transmission of AGPs into the environment and their possible entry into the human body by microorganisms will be also valuable information in future.

Response:

We disagree with your suggestion of conducting a statistical analysis for each week separately to compile individual p values by week. We analyzed the growth performance data from each of these experiments as a complete trial using repeated measures as opposed to doing a separate anova for each time point. Repeated measures is appropriate here because the same animals are being weighed every week, just at different time points. By using repeated measures we are able to recognize that the animals weight at the first time point will impact their weight at the following time point and so on. If the analysis were done on all these timepoints separately, that is assuming that each time point is independent from the next, it would potentially bias the results because we know early growth is going to impact growth later in the nursery phase. When doing a repeated measures analysis, all the time points get evaluated at once and only produce one P value to determine if the treatment had a significant difference in growth for the entire period, not just a single timepoint. Time is included in the model, but we are only running one model and not a separate model for each time point. If there is only a difference in growth during one period, we will get a treatment by time interaction (as observed in the BW data in this manuscript). Using this interaction and subscripts when differences are present helps us compare results within each time point, but when using repeated measures we will not get a p value for each specific time point. To get comparisons between treatments we used a post hoc tukey test to create subscripts as opposed to linear or quadratic contrasts. These subscripts are used to differentiate significant differences at P < 0.05. Please refer to this reference, which was previously provided, to gain a better understanding of using repeated measure in statistical analysis as we have correctly done (Statistical analysis of repeated measures data using SAS procedures. RC Little, PR Henry, and CB Ammerman. Journal of animal science, 1998. 76:1216-1231).

Reviewer 4 Report (New Reviewer)

Comments and Suggestions for Authors

Please provide the diet formulation for the negative control group.

Comments on the Quality of English Language

Minor editing of English language required

Author Response

Please provide the diet formulation for the negative control group.

The exact diet formulation is proprietary information, but more information was added to the methods section to describe the diets to the reader on line 151.

This manuscript is a resubmission of an earlier submission. The following is a list of the peer review reports and author responses from that submission.

Round 1

Reviewer 1 Report

Comments and Suggestions for Authors

The manuscript presents interesting results on a recent topic. It is well written and explored. The authors provide important questions and answers during the discussion. Besides a few corrections, I reccommend the manuscript for publication.

Abstract

The authors need to add some of the numerical results and p values hen talking about differences or lack of.

A conclusion should also be added.

L61: Superscript 4. Not sure if it is a mistake or if it is intentional.

Discussion

Same as the abstract, the authors should add p-values hen ritting about statistical differences.

Author Response

The manuscript presents interesting results on a recent topic. It is well written and explored. The authors provide important questions and answers during the discussion. Besides a few corrections, I reccommend the manuscript for publication.

Response: Thank you.

Abstract

The authors need to add some of the numerical results and p values when talking about differences or lack of.

Response: A P value was added to the abstract (line 42) to indicate a significant difference. Non-significant P values were not added to the abstract due to space limitations and the abundance of comparisons made. However, non-signficant P values were added to the discussion section where more elaboration was appropriate.

A conclusion should also be added.

Response: The final statement was revised to serve as a conclusion (line 43-45).

L61: Superscript 4. Not sure if it is a mistake or if it is intentional.

Response: the superscript “4” was deleted and replaced with [4].

Discussion

Same as the abstract, the authors should add p-values hen writting about statistical differences.

Response: P values were added for both significant and non-significant differences.

Reviewer 2 Report

Comments and Suggestions for Authors

This is a well written paper and the authors should be commended for their presentation of a series of three studies that appear to have been conducted under well-established protocols and with careful attention to measuring parameters in pigs that are consistent with published literature when feeding similar feed additives.

However, after careful examination the primary concerns with this paper focus on the levels of the feed additives being tested and whether or not they could have ever been expected to elicit a measurable response in pigs. The authors note that they used manufacturer's recommendations for formulation of the diets, yet they can clearly cite published papers (in the case of the orange extracts) that would indicate that their added levels were much lower than studies where effects were seen. One of the key stated objectives of the paper is to "determine potential biological mechanisms" (Line 19 and 30-32) for improved growth when fed to pigs, yet the experimental diets were clearly not designed to obtain this result.  The diets appear to be best suited to simply test the products' recommended feeding levels, yet readers are not able to know what products were used - which could be useful for researchers in this field if they were wanting to use similar products themselves. It was also noted by the authors that pigs were not placed into an environment or challenged in ways that could also have demonstrated the value of these additives.

In short, if the specific products being tested are reported then this data becomes useful for further work with these compounds - as it shows that very little effect comes from the feeding levels described here - but without that key information this data doesn't contribute novel findings or validate past work in this area.

Author Response

This is a well written paper and the authors should be commended for their presentation of a series of three studies that appear to have been conducted under well-established protocols and with careful attention to measuring parameters in pigs that are consistent with published literature when feeding similar feed additives.

Response: Thank you.

However, after careful examination the primary concerns with this paper focus on the levels of the feed additives being tested and whether or not they could have ever been expected to elicit a measurable response in pigs. The authors note that they used manufacturer's recommendations for formulation of the diets, yet they can clearly cite published papers (in the case of the orange extracts) that would indicate that their added levels were much lower than studies where effects were seen. One of the key stated objectives of the paper is to "determine potential biological mechanisms" (Line 19 and 30-32) for improved growth when fed to pigs, yet the experimental diets were clearly not designed to obtain this result.  The diets appear to be best suited to simply test the products' recommended feeding levels, yet readers are not able to know what products were used - which could be useful for researchers in this field if they were wanting to use similar products themselves. It was also noted by the authors that pigs were not placed into an environment or challenged in ways that could also have demonstrated the value of these additives.

Response: The feed additives tested were proprietary and the inclusion rates recommended by the manufacturers were those we used. As indicated, there were very few published studies evaluating these products in weaned pig diets to provide additional guidance on diet inclusion rates that may elicit growth improvements. We speculate that the high health conditions of pigs and their environment used in these experiments may have led to the lack of growth improvements observed, but it may also be due to the lack of effectiveness of the amount or chemical form of potentially biologically active components in the additives.

In short, if the specific products being tested are reported then this data becomes useful for further work with these compounds - as it shows that very little effect comes from the feeding levels described here - but without that key information this data doesn't contribute novel findings or validate past work in this area.

Response: We believe that the results of this study do provide valuable insights of the expected (or lack thereof) growth, metabolome, and microbiome responses from feeding these various types of additives to weaned pigs.

Reviewer 3 Report

Comments and Suggestions for Authors

The manuscript proposes to describe simultaneously 3 trials which they were designed to explore the efficacy of different additives. A negative and positive control are included in each trial in order to test the efficacy of additives compared to them. Each one the trials would allow the preparation of an independent report about their efficacy, as they are already described in different repeated tables. However, merging the 3 implies the design of a very large manuscript with many tables with not significant results. There is no an additional value of merging tables from different experiments. The lack of responses with the additives don´t allow to perform an horizontal discussion about the relevance of variables.

Experiments performed in good experimental conditions confirm that it is not a good idea to evaluate additives in these conditions, but after a tremendous effort to perform the trials and to read so long paper, the feeling is that there are not new results or novelty to the existing literature.  

On the read I have also detected the need to change the first row of tables 2 and 3 to identify experiment 2 and 3.

I would suggest to delete some tables that they don´t provide interesting results (Table 5, 6, 7), or Aas content from Tables 8-10. 

Simplify microbiota taxons description in Table 11

Author Response

c

The manuscript proposes to describe simultaneously 3 trials which they were designed to explore the efficacy of different additives. A negative and positive control are included in each trial in order to test the efficacy of additives compared to them. Each one the trials would allow the preparation of an independent report about their efficacy, as they are already described in different repeated tables. However, merging the 3 implies the design of a very large manuscript with many tables with not significant results. There is no an additional value of merging tables from different experiments. The lack of responses with the additives don´t allow to perform an horizontal discussion about the relevance of variables.

Response: We agree that there are many tables showing no differences among treatments for many variables measured in these 3 experiments. Therefore, we have moved many of the tables showing minimal or no significant differences among treatments to the supplemental data document to reduce the length of the manuscript.

Experiments performed in good experimental conditions confirm that it is not a good idea to evaluate additives in these conditions, but after a tremendous effort to perform the trials and to read so long paper, the feeling is that there are not new results or novelty to the existing literature.  

Response: We believe that there is value in reporting lack of growth, metabolome, and microbiome responses from feeding these additives to weaned pigs.

On the read I have also detected the need to change the first row of tables 2 and 3 to identify experiment 2 and 3.

Response: Corrections were made.

I would suggest to delete some tables that they don´t provide interesting results (Table 5, 6, 7), or Aas content from Tables 8-10. 

Response: Tables 5, 6, 7, 8, 9, and 10 were moved to supplementary material to reduce the amount of non-significant results in the main manuscript. The manuscript text was also revised accordingly.

Simplify microbiota taxons description in Table 11

Response: Table was revised to include only genus and species.

Reviewer 4 Report

Comments and Suggestions for Authors

Very nice job in presenting your work and the experimental design is very good as well. But please consider the following:

1. Try to use the same font type and size (specially in the tables, they have different font compared to the text)

Author Response

Very nice job in presenting your work and the experimental design is very good as well. But please consider the following:

  1. Try to use the same font type and size (specially in the tables, they have different font compared to the text)

Response: The table and figure fonts were revised to match the font size of the text in the manuscript as suggested.

Round 2

Reviewer 2 Report

Comments and Suggestions for Authors

After reading the revised paper, the changes to the manuscript made by the authors do help to clarify the connections between the 3 studies and reorganization of the tables helps to keep the focus on the results that do have impact on readers/scientists working in this area. Ultimately, my concerns with the paper still exist, yet the strength of the peer review process lies in the input from multiple sources and I see no other reasons why publication of this manuscript shouldn't proceed.